# The Revolution of Animal Genomics in Forensic Sciences

**DOI:** 10.3390/ijms24108821

**Published:** 2023-05-16

**Authors:** Irene Cardinali, Domenico Tancredi, Hovirag Lancioni

**Affiliations:** Department of Chemistry, Biology and Biotechnology, University of Perugia, 06123 Perugia, Italy

**Keywords:** animal genomics, domestic, wildlife forensics, canine DNA, genetic markers

## Abstract

Nowadays, the coexistence between humans and domestic animals (especially dogs and cats) has become a common scenario of daily life. Consequently, during a forensic investigation in civil or criminal cases, the biological material from a domestic animal could be considered “evidence” by law enforcement agencies. Animal genomics offers an important contribution in attacks and episodes of property destruction or in a crime scene where the non-human biological material is linked to the victim or perpetrator. However, only a few animal genetics laboratories in the world are able to carry out a valid forensic analysis, adhering to standards and guidelines that ensure the admissibility of data before a court of law. Today, forensic sciences focus on animal genetics considering all domestic species through the analysis of STRs (short tandem repeats) and autosomal and mitochondrial DNA SNPs (single nucleotide polymorphisms). However, the application of these molecular markers to wildlife seems to have gradually gained a strong relevance, aiming to tackle illegal traffic, avoid the loss of biodiversity, and protect endangered species. The development of third-generation sequencing technologies has glimmered new possibilities by bringing “the laboratory into the field”, with a reduction of both the enormous cost management of samples and the degradation of the biological material.

## 1. Introduction

Genomics plays a pivotal role in forensic science. During a forensic investigation, the collection of biological material such as saliva, seminal fluid, or hair is routine. Nowadays, to improve the quality of forensic reconstructions, law enforcement agencies take non-human biological samples as “evidence” [1]. Often domestic animals represent an important forensic tool for the identification and detection of animal attacks or episodes of property destruction [2]. Further applications also include food contamination and frauds [3], botany [4,5], microbiomics [6], civil and criminal investigations on animal theft and abuse, and the individuation of a link between victims, perpetrators, and/or crime scenes [7]. However, regardless of the relevance that non-human biological samples may have within a crime scene, only a few laboratories in the world are able to carry out a valid forensic analysis of these samples. In order to ensure data admissibility before a court of law, they must adhere to the standards and guidelines of a crime laboratory [8] and undergo the approval of a specialized ethics regulatory board for sampling living animals [9]. The lack of representative data and deep knowledge of species testing protocols, and the key role of guidelines interpretation represent the main hurdle that laboratories have to pass to submit correct results. For this reason, scientific and technical capabilities of laboratories and their personnel should be continuously evaluated to improve the overall quality of both animal forensic genetic testing [10] and molecular, evolutionary, and phylogenetic knowledge about the species considered [11].

Due to the standardization and the accreditation of processing procedures, laboratory protocols, statistical models, and techniques, human forensic genetic analysis currently represents the “gold standard” by which other forensic sciences are measured [12]. To reduce the ambiguity of collaborations between laboratories skilled in animal genetics and law enforcement agencies within a forensic investigation, in 2008, during the International Society for Animal Genetics (ISAG; the Animal Forensic Genetics Standing Committee) conference, members of the committee defined animal forensic genetics as “the application of genetic techniques and theories relevant to legal or law enforcement issues concerning animal biological material”. Recently, different genetic tools have been developed to improve the routine forensic procedures that need the identification of mammalian species [13]. In animal or human criminal investigations, the forensic genomics analysis of animal biological materials considers both domestic and wildlife [11]. When traces of a domestic species’ biological material are relevant, the identification of the species is less important than the accurate and correct individuation of the “subject”. In this scenario, the application of forensic sciences to wildlife is also gradually gaining relevance since there is a growing refusal of illegal trafficking [14,15], which has disastrous effects on the conservation of species and biodiversity [16,17].

In forensic sciences, the establishment of reference databases and platforms helps to carry out studies focused on genetics and conservation biology. As for humans, there are many sources that forensic scientists and scholars can refer to for the analysis of animal species and a better sample localization of rare and endangered species [18] (Table 1).

The collection, storage, and use of DNA information and any other data must follow the ethical, legal, and social regulations of each country and should be continuously updated since they represent one of the most important investigative resources in contemporary criminal justice systems.

## 2. Animal DNA in Forensic Cases

The employment of biological material from a domestic animal in forensic investigations is becoming frequent [19,20]. One of the first cases in which the analysis of animal DNA has been applied to solve a legal case concerns an episode of illegal cattle purchase that occurred in the Western Pyrenees between France and Spain. Six samples were used for genetic analyses: three belonged to alleged stolen animals, and three to alleged mothers. The analysis of four gene loci produced maternal matches for each sample analyzed and was considered sufficient evidence that animals were stolen [21]. However, the most famous case of forensic employment of animal DNA affects Snowball cat, used to solve a human-on-human crime. In 1994, the corpse of a woman was found in eastern Canada, with a bloody leather jacket that was used to cover the body. The main suspect in that murder was the victim’s husband, but the blood found at the crime scene was attributed to the murdered woman, while the jacket ownership was difficult to be defined. The analysis of this jacket revealed the presence of hair, later attributed to a cat. Although the suspect did not have it, his parents owned a white cat named Snowball. Consequently, a blood sample from that cat was analyzed and compared to the hair sample found at the crime scene. Samples perfectly matched, thus unequivocally correlating the man with his wife’s death [22].

Since 2012, when a murder was solved for the first time in the Winchester Crown Court, thanks to the mtDNA profile of Tinker, the suspect’s pet cat, the importance of cat fur to crack forensic cases was clear. Due to their clingy hair as well as their habit of licking themselves for grooming, cats are particularly well-suited for forensic analyses [23,24,25]. Recently, Monkman and colleagues focused on the transfer of human genetic material to the household cat through patting or brushing and its persistence on their hair, thus providing a preliminary study highlighting its relevance to the sample targeting [26].

Another example of the involvement of pets in solving crimes concerns a British bouncer stabbed to death outside the club where he worked in the early 2000s. A trail of blood leading away from the body was analyzed, and it was found to be of canine origin. The main suspect was a man who had previously been removed from the club and owned a dog. Blood samples and dog references were sent to the Veterinary Genetics Laboratory at the University of California for molecular analyses, and the STR data produced a match with the suspect man’s dog profile. The conclusion of the investigation revealed that, after being denied entry, the man stabbed the bouncer, also injuring his own dog [21].

In some cases, the same methodological and molecular approach seen for the British bouncer was used to analyze the mitochondrial DNA (mtDNA) since nuclear DNA was found to be ineffective. In the 2002 Westerfield vs. Superior Court case, a seven-year-old girl was abducted and murdered. A main piece of evidence in the case was a sample of dog hair found in the mouth of the suspect’s dryer: investigators were aware that the girl owned a pet, while the man suspect did not. A hair without a root usually contains little nuclear DNA; thus, in that case, it was not possible to generate a nuclear DNA profile. Therefore mtDNA was analyzed, and its classification in one haplotype typical of only 9% of the world’s canine population, including the victim’s dog, was used as evidence to link Westerfield to the murder of the little girl and to obtain his conviction [21]. Additionally, there are other examples that could testify to the advantages of analyzing animal genomes to solve a crime [27,28]. However, forensic genetic analysis often also involves livestock farms with a large application on the establishment of a confident parentage and high traceability across the agri-food sector [29,30,31,32,33,34] and small mammals such as ferrets, hamsters, and rabbits which are becoming more popular as companion animals ([35,36,37,38] as examples).

On the other hand, there are many forensic cases of abuse of animals by humans, which intentionally injure, kill, or trade pets, farm animals, or wildlife without leaving a trace. Therefore, the investigation could involve the analysis of damaged biological materials that need a particular optimization [39]. Unfortunately, cases of animal sexual abuse (ASA) have also been increasingly reported and defined as the eroticization of violence, control, and exploitation [40]. They encompass a broad spectrum of behaviors and affect many species [40,41,42,43].

## 3. Genetic Markers Used in the Forensic Field

Nowadays, forensic science has become a highly interdisciplinary science since it uses concepts and methodologies from dactyloscopy, traceology, molecular biology, veterinary forensics, and computer forensics [44]. Over the years, there was a gradual expansion of analysis to include different protein polymorphisms, and in the 1970s, electrophoresis of proteins such as plasminogen or transferrin became the dominant methodology. Subsequently, in the second half of 1980, researchers carried out non-coding fragments of the myoglobin gene showing that these were highly polymorphic between individuals; this observation led to the birth of a new branch, forensic genetics [44]. It is based on two rules: firstly, a single cell with a nucleus contains all the genetic information of an organism. The second rule, on which all branches of criminology are based, is Locard’s exchange principle: “contact between two objects leads to an exchange of substances between them, so that the perpetrator both takes material away from the scene of the incident and leaves traces of his or her presence” [44]. Consequently, the biological matter “left behind”, containing the complete information about an organism, enables the identification of an individual. Over the past three decades, many genetic markers have been used to improve DNA databases [45]. The first was minisatellites, as the difference in length of these repeated sequences between individuals provided the basis for the use of polymorphisms in identification testing. This approach was developed by Jeffreys and colleagues, who analyzed the hypervariable regions of introns in the human myoglobin gene, demonstrating the presence of highly variable regions in nuclear DNA [44,46]. The analysis of minisatellite sequences combined with hybridization allows for obtaining an individual genetic fingerprint. Jeffreys was undoubtedly a pioneer in forensic genetics, having been the first to use molecular identification for forensic purposes in the famous Pitchfork case, in which Colin Pitchfork was the first person to be convicted on a genetic basis [47]. More recently, microsatellites, also known as short tandem repeats (STRs), are preferred to minisatellites due to their advantage in the analysis of degraded material [44,48] and their outstanding discriminatory power that provides a virtually unique combination [49,50]. STR markers were first used in the early 1990s; the first multiplex system (QUAD) capable of amplifying four STR markers was designed by the Forensic Science Service in the United Kingdom and, in 1994, was introduced into forensic practices. Microsatellites gained popularity due to their high level of polymorphism, producing results that constituted strong evidence [44,50] and their capability to predict the phenotype [51,52].

Variants of short repetitive DNA sequences are observed in both sex chromosomes, Y-STRs and X-STRs, and autosomes. Autosomal STRs allow individual identification, while sex STRs are useful for determining sex [44]. Given the uniparental inheritance of the Y chromosome, the Y-STR profile of related individuals turns out to be identical, and when combined with the analysis of familial searching and an investigative genetic genealogy, it allows the identification of donor relatives [53]. Consequently, a suspect group can be narrowed down to individuals who share the same Y chromosome haplotype, and among them, the individual whose autosomal profile matches the evidence can be identified as a possible suspect [44]. Even if human samples are largely overwhelming in genetic casework, STR polymorphisms are also employed to analyze animals, such as in cases of poaching, illegal trafficking of protected species, and mistreatment [54], or simply to determine whether two analyzed samples derive from the same individual [44]. There are many STR typing systems used to identify animal species that have to be developed and standardized for a vast number of species [55]; thus, STRs are increasingly being joined by other types of DNA polymorphisms. Although single nucleotide polymorphisms (SNPs) have a smaller discriminatory power, they represent an additional tool to complement STRs rather than a replacement, useful in the analysis of degraded biological material [44].

SNP markers are successfully used in the analysis of samples from disaster victims or archaeological/paleontological remains, which contain a small amount of degraded genetic material. Analysis of SNPs in biological traces often involves mitochondrial DNA (mtDNA) when nuclear DNA (nDNA) is not available or exploitable. Due to greater resistance to degradation and the presence of numerous copies per cell, mitochondrial DNA can be successfully analyzed in biological samples subjected to physical and chemical degradation from fossil bones or lost hair post-mortem [56,57]. In the human mitogenome, SNP analysis is performed both on coding and non-coding fragments and was proved useful in forensic cases providing additional information to support poor or negative autosomal information (reviewed in [58,59]).

In accordance with guidelines concerning mtDNA typing issued by the DNA Commission of the International Society of Forensic Genetics, the entire control region must be sequenced for maximum data reliability [60]. Nowadays, the extensive mtDNA typing of population data and the establishment of dedicated databases such as EMPOP (www.empop.org) provide as many as possible mitogenomes at a maximum level of resolution and quality. Phylogeographic investigations conducted on local populations distributed worldwide ([61,62,63,64] as examples) constantly contribute toward the definition of reference population databases to be used at a microgeographic level. A branch of forensic genetics that frequently uses mitochondrial polymorphisms in conjunction with DNA barcoding is wildlife forensics, which deals with crimes perpetrated against protected species. In this case, the identification of a species is accomplished by the analysis of the cytochrome b gene, which is extremely conserved within the same species [65]. It should be further emphasized that SNPs play a relevant role in forensic genetics, not only for the identification of individuals but also for reconstructing the phenotypic profile of a subject from its DNA sample [66,67,68,69].

The development of genetic markers also finds an important application in the analysis of laboratory animals by ensuring the purity of breeding lines, good monitoring of model species, and the identification and variation of genes acting on specific traits [70,71,72,73,74].

## 4. Recommendations for Animal DNA Analysis in Forensic Sciences

In the analysis of genetic material from an animal source, the identification of a species and an individual represents the objectives to be achieved.

The most used loci in species testing are the mitochondrial cytochrome b (cyt b), which are the preferred gene locus for taxonomic analysis and species identification, and cytochrome oxidase subunit 1 (COI), for which the sequencing of 600 base pairs (bp) was proposed as a method to record the terrestrial biodiversity [75]. When differentiation between closely related species is required, the analysis of the mtDNA D-loop (control region; CR) will be more appropriate [76]. Additionally, the use of a single genetic marker cannot produce high levels of confidence in taxonomic identification [75]. Investigations in which the victim is an animal or the crime is the illegal trafficking of protected species may be subordinate to investigations involving human victims. However, sample collection and preservation follow the same standards as in any forensic investigation. The integrity and the traceability of evidence require particular attention to ensure the normal continuation of the ongoing court case. In this context, thirteen recommendations, representing the first guidelines for the standardization of animal DNA typing in forensic investigations, should be considered [75] in addition to standards and guidelines that regulate the entire process of DNA analyses from both domestic and wildlife samples [77,78,79] (Table 2).

The development of a solid genotyping system for animals is one of the main goals of forensic scientists, which aims to reach quality standards of human genetic testing. When comparing genetic markers used in animal (Table 2) and human (Table 3) DNA analysis, some differences could be observed.

Many standardized commercial kits for human DNA testing allow for the same achievement results in any laboratory performing the analysis with different equipment [8], even if, in some cases, the inter-laboratory interpretation of results could lead to different evaluations [60].

In animal forensic genetics, the lack of representative data and high-quality genomic sequences for many species ensures a continued assessment of laboratory capabilities to improve the overall quality of genetic testing [8,10].

STR profiling represents the gold standard both for animal and human genetic analyses since it has a high discrimination power among individuals [8,78,79,81], but when the genomic material is highly degraded, standardized kits with shorter versions of STRs are employed to overcome this challenge [8]. Loci showing two-base repeats, however, suffer from increased stutter and an altered balance of heterozygosity. Consequently, the use of dinucleotide repeats in animal forensic genetics is discouraged, except for those markers that have already been extensively used in animal genetics studies [75].

To resolve kinship structures and determine the sex, variants of short repetitive DNA sequences in sex chromosomes are the preferred genetic markers; unfortunately, they have a great limitation in analyses concerning animals, as they are uncharacterized for many species [78]. When X- or Y-STR typing is used to establish a relationship between individuals within a forensic analysis, the probability of mutations and the linkage must be considered [75,84]. In the case of a genetic inconsistency, the probability of mutation must be reported [85]. Furthermore, a reasonable number of samples must be considered to estimate allele frequencies, and individuals must be representative of the population from which the unknown sample might come. Often a cluster of 200 individuals from a representative population is considered a de facto standard [86], although the sample size depends on the number of potential contributors and the level of locus diversity. The frequency databases should be analyzed for Hardy–Weinberg equilibrium, and any deviation from that equilibrium should be noted [75]. A widely used factor in forensic genetics influencing the probability of sharing an allele between two individuals with a common genetic ancestor is the kinship factor, also called “Fst”. In the modern human population, the degree of common ancestry is typically low (between 0.01 and 0.03), contrary to animal populations, which present a higher Fst value. Kinship factor values of several wild animals, such as deer, bears, and European badgers, have been published and reflect the amount of inbreeding within these wild populations [75]. Regarding the presentation of produced data, the format of each forensic report depends on the criminal justice system and must state the purpose and principal of the investigation. Moreover, laboratories performing routine analysis of animal DNA for forensic purposes should consider the accreditation to ISO17025, as recommended by the International Society for Forensic Genetics (ISFG) [87].

Since many samples addressed to forensic purposes belong to a human/animal mixture, laboratory analyses could provide aspecific genomic products due to inter-species contamination [88]. In this context, a standardization for animals is crucial to discriminate non-human DNA. To perform validating tests, the biological material of a reference sample must come from a known source (i.e., zoological institutes), but in the absence of sample specimens, the authentication of species by sequencing a gene locus (cyt b or COI) [89] compared to sequences in a database is equally acceptable. For molecular analyses, many primer sequences (universal or species-specific) for various taxa have been published. When using universal primers, a mix of DNA from different species must be considered. In the case of loci located on mitochondrial DNA, the complete genome sequence is preferred, but default sequences of mtDNA CR, i.e., for dogs, have been developed [90] and serve as a model for future mitochondrial comparisons [91]. The species specificity must be demonstrated for each primer by providing new data or referring to previously published data. For these tests, not only the most common species should be considered, but also any species phylogenetically closely related to the species under analysis [75].

## 5. Domestic Animal DNA as Forensic Evidence

Animal evidence represents an important tool to be analyzed when an association between a crime scene and a suspect has to be established or a crime involves a specific animal. In this context, the most studied species is the domestic dog, which represents an undisputed faithful companion of humans. Canine evidence, such as hair samples, can act as a link between the victim and a hypothetical crime suspect because of the close physical contact between the dog and owner [92]. Usually, dog or cat hair is associated with forensic evidence such as clothing, car seats, and home furnishings [17]. D’Andrea and colleagues demonstrated that it is almost impossible to enter a pet owner’s home and not be “contaminated” by animal hair [91]. In addition, since dogs have more hair than humans, the crime scene can contain more hair from dogs than humans, and the probability of finding canine-derived hair is higher compared to recovery traces of human biological material. The detection procedure for animal DNA from a crime scene, as for human DNA, is based on the analysis of STR markers ([93,94] as examples). However, STR profiling helps to understand whether a given dog is responsible for an attack or whether a hair sample obtained from a crime scene can be attributed to a specific dog (owned by a possible suspect). In some cases, the canine DNA sample does not match the suspected dog, or no reference genetic material is available; thus, the evaluation of morphological information (coat color, body size, etc.) from eyewitness statements is very useful. Nevertheless, since eyewitness testimonies are frequently absent or extremely subjective and unreliable, data obtained from eyewitness accounts have to be supplemented with phenotypic characteristics deduced from the identified canine DNA profile [69,92]. DNA analysis, especially when accompanied by canine DNA phenotyping, can bypass these limitations by allowing a reliable reconstruction of the subject appearance.

### 5.1. Predicting Canine Phenotypic Traits: Canine DNA Phenotyping

A recent study conducted by Berger and colleagues aimed to develop a molecular genetic tool capable of predicting the outwardly observable characteristics of dogs from DNA samples, to be applied in forensic contexts [92]. Domestic dogs underwent strong human selective pressure, which led to a wide diversity of morphologic traits and behaviors among breeds [95,96,97]. An accurate phenotypic description of a canine individual through the analysis of many genetic markers has been proposed for different morphological traits (body size, coat color, pattern, and structure; bulk; skull shape; ear shape; limb length; digit type; tail morphology) [69,92].

The validation for the employment of different genetic markers in canine DNA phenotyping stimulated the world scientific community to further investigate the topic in different forensic fields [98] by searching for more reliable markers and expanding the sample cluster. Combining the analysis of different genetic markers (STRs, SNPs, and indels) could allow the prediction of breed [99] or phenotype with more accuracy (as reviewed in [69]). Moreover, all these recent statements on genotype–phenotype relationships in dogs cannot exclude applications that are also in the evolutionary research field, such as in the analysis of genetic molecular mechanisms involved in the domestication process.

### 5.2. Fecal Samples as a Source of Animal DNA

Biological material from dogs, such as hair, saliva, or feces, may provide a useful source of information, although sometimes with difficulty. DNA extracted from a fecal sample often appears highly degraded due to environmental factors and the continuous deterioration operated by a multitude of bacteria present in the intestinal tract [100]. In addition, fecal material contains many compounds known to be inhibitors of one of the most used techniques in molecular biology, the polymerase chain reaction (PCR) [101]. Since receiving fecal samples is not so common in a forensic laboratory and producing a genotypic profile from such a degraded DNA sample is a challenge, scholars agreed to analyze a canine feces sample from a crime scene to solve a robbery-murder case [100].

A dog feces sample (reference) was collected from the house where the crime occurred; a suspect was arrested, and another fecal sample was collected from a man’s shoes (evidence) to link the latter to the crime scene. Then, the DNA was extracted from cells located on the surface of feces and sequenced for hypervariable fragments of the mitochondrial control region. Although mtDNA has a less discriminating power than STR or nDNA SNPs, the high copy number per cell also makes it an excellent tool in case of small and/or degraded amounts of DNA [100]. Then, to determine whether the two stool samples belonged to the same dog, 15 canine STRs belonging to the ISFG standardized and recommended list for canine genetic identification and two fragments (800 bp and 145 bp long, respectively) of the mtDNA control region were analyzed. As expected, the STR panel genotyping did not show positive results, probably due to the low quality of the extracted canine DNA, while the second fragment of the mtDNA control region could be amplified and sequenced, thus confirming the usefulness of this mitochondrial region in typing degraded and low-quantity samples. The resulting alignment showed that the DNA sequences obtained from the fecal sample matched with those of the species *Canis lupus familiaris* (domestic dog), consequently excluding any other species as a source of forensic evidence.

After the comparison with 23 described canine haplotypes, both the evidence (fecal sample collected from the suspect’s shoes) and the reference (canine fecal sample collected at home) showed a 100% identity with a specific mtDNA control region haplotype [100]. This study showed three important pieces of evidence: (1) The analysis of animal-derived biological material turns out to be increasingly crucial for the positive turnaround of many investigations. (2) The mtDNA control region is an excellent tool for typing DNA extracted from degraded and low-quantity sources. (3) Fecal material, although difficult to be analyzed, is a powerful source of information for different applications.

### 5.3. The Dog as Suspect

As extensively mentioned, pets, especially dogs, have proven to be faithful companions of humans in everyday life, but sometimes this relationship does not have positive consequences. Fatal attacks by dogs have been sporadically reported, and deaths are often the result of unwitnessed attacks [102]. Dogs typically drag prey to the ground and then maul it, attempting to ‘disarm’ the victim by striking its limbs. Once the victim has been knocked down, the animal usually begins to bite its throat, the back of its head, and the cranium; if the attack continues, the victim will die from asphyxia, hemorrhaging, or cranial fracture and its complications [103]. Unlike cats, dogs eat their prey before it dies if their intention is to kill. When dogs live in packs, each member tears the prey, trying to eat as much food as possible [104]. Attacks by domestic dogs usually aim to head and neck regions, whereas those by stray dogs often affect hands and legs, probably due to the different human behavior towards domestic and wild dogs [103]. Children are usually bitten in the head and nape since these anatomical regions are ‘at dog height’ [103]. During a defensive or anger bite, a dog may attempt to bite and run away, producing a bite mark with the anterior region of the dental arch (mainly incisors but also canines), very similar to a human bite. Conversely, during a predatory bite, a dog may forcefully grip a flap of tissue using the posterior dentition (mainly carnassial) and cause severe lacerations by shaking and pulling. If the dog then loses its grip, it will move its head forward, trying to grab more tissue, thus producing a multitude of superimposed bite marks [103].

The extent of damage produced by a dog attack depends on the vulnerability of the victim: children, the elderly, and disabled people have the highest mortality rates due to their lack of strength, indefensibility, and reduced body size [103]. When dogs are part of a pack, ‘pack instincts’ tend to intensify the attack [104,105]; therefore, the threat will be proportional to the number and size of dogs in the pack.

When a dog bite does not turn out to be fatal, forensic investigations are usually conducted without trouble as the victim can testify and reconstruct the attacker’s identity; however, it may happen that no witnesses are present, and this makes the reconstruction more complex. During the investigation, all evidence must be stored, and all actions have to be noted; all observed injuries (especially puncture wounds) should be photographed and, if possible, swabbed with dog-specific kits [103]. The collection of biological samples from wounds is also recommended [106]. The forensic approach for a suspected dog attack should include a detailed analysis of the victim and dog and a thorough evaluation of the crime scene.

Four steps are recommended: (1) obtaining information regarding the circumstances, witness statements, age of the victim, dog breed, etc.; (2) careful observation of footprints, environmental conditions, the position of the corpse, etc.; (3) collecting biological traces, blood samples, hair, textile fibers, etc.; (4) producing documentation in the form of photos, diagrams, drawings, etc. If the dog’s identity is known, it is possible to also analyze its past behavioral patterns [102,103,107]. To collect biological samples from the suspected dog, the animal could be anesthetized, and emetics may be administered to induce vomiting in order to examine its gastric contents. The presence of commercial food suggests that the dog is domestic (and therefore not starved during the attack), while the presence of game suggests a wild dog [103]. Jewelry, rings, textile fibers, and bone fragments are also frequently found in canine feces; thus, a radiographic investigation is strongly recommended [103,108]. In addition, a toxicological assessment of its fluids and tissues should be performed to dispel any doubts about the possible administration of excitants or steroids that could have altered its behavior [102].

When the identity of a dog is unknown, two types of analyses are used for the identification: odontological and genetic. Since the size and damage patterns of a bite mark are characteristic of each breed, the first methodology provides macroscopic discrimination among different suspected individuals by excluding dog breeds and identifying a potential one. The genetic approach is based on the analysis of canine DNA extracted from a sample of biological material retrieved from the lesion [109]. Although this approach represents a valid practice in forensics, it does not lack limitations: the collection of saliva samples from the victim’s wound could take place in a variable time interval and, since the first action carried out on the victim is an aid, the use of disinfectants on tissues can drastically ‘erase’ most of the biological traces produced by the attacker. In this regard, Iarussi and colleagues have recently proposed a new and more qualitative approach based on searching for the victim’s genetic profile in the mouth of the suspected dog instead of focusing on the wound produced on the victim [109]. They examined ten dogs of different breeds and had them bite into a piece of beef, with the aim of finding out whether it was possible to trace the genetic profile of the bovine sample using buccal swabs. Their results proved to be very promising since traces of bovine DNA from the buccal swabs were found, and an inverse proportionality relationship between the extent of the time interval after the bite and the quality of DNA sampling was shown, thus increasing the importance of this innovative approach to improving the reliability of forensic evidence produced by genetic laboratories [109].

## 6. Wildlife in Forensic Genetics

A branch of conservation genetics that has been recently receiving much attention is the application of analytical techniques to produce DNA evidence that can support the enforcement of legal conservation regulations, commonly referred to as “wildlife DNA forensics” [110]. It concerns the identification of species, populations, phylogenetic relationships, and/or individual identity of a sample, and it represents a specialized area between wildlife conservation research and applied forensic science [111]. Forensic genetics is becoming the key to fighting illegal wildlife trafficking [18] and establishing national and international legislation to protect the biodiversity of habitats and species [15,16]. This is also deeply intertwined with the protection of the legal economic chain, passing through the meticulous control of biologically derived products, to identify the wild meat belonging to a particular or protected species [112,113] and control the correct food composition in raw materials such as cheese or meat [114].

The international trade in endangered species is regulated and monitored by the Convention of International Trade in Endangered Species of Wild Fauna and Flora (CITES), which stipulates a list of protected species, thus stimulating international research to develop methods for species identification [17].

Many countries have judicial systems that punish the collection in the wild, breeding, ownership, or international trade of animal parts or whole organisms. The US Endangered Species Act (ESA) of 1973 is a key piece of domestic (US) and international animal conservation legislation; it provides a legislative framework for the protection of endangered species and their habitats. In America, the Lacey Act is central to the protection and conservation of wildlife, and at the time of its enactment (1900), it was the first federal law to protect wildlife. The Lacey Act covers all fishes, wildlife, their parts or products, and plants protected by CITES and those protected by state laws, thereby regulating the international trade in protected species. In Europe, the EU Council Regulation (EC) No 338/97 regulates the protection of species of flora and fauna by monitoring the trade in any species included in the regulation. It lays down provisions for the import, export, and re-import, as well as trade within the EU, of specimens of listed species [17]. In this context, the aim of forensic analyses is to provide information or evaluate hypotheses related to the available evidence. However, in wildlife law enforcement, most questions address the identification of species, geographic origin, individuals, and families of each sample [111] (Figure 1).

### 6.1. Species Identification

The most frequent forensic application of genetic analysis in wildlife DNA is related to the identification of species, especially to fight against poaching [115]. It is based on the isolation and the analysis of genetic markers that show variation between species but are generally conserved within species. In animals, the most used markers are the mtDNA cytochrome b and cytochrome oxidase subunit 1 (COI) since the mutation rate of the latter coincides strongly with the rate of evolution of the species [111]. When only scant biological material is available, the use of mtDNA increases the success of the analysis [13].

At this level, interspecies differences are largely due to changes in single base pairs in the DNA sequence, known as single nucleotide polymorphisms (SNPs).

SNPs can be directly employed to identify animal species without sequencing the entire genome, but only using techniques such as restriction fragment length polymorphism PCR (RFLP) or SNP genotyping [111]. Minimizing the size of targeted genetic markers is often necessary to obtain reliable results from samples that have been both degraded and highly processed, producing DNA fragmentation [1,116]. Both PCR-RFLP and SNP genotyping are widely accepted in the forensic community, although these techniques are often applied for species detection rather than species identification. Indeed, the application of these techniques requires knowledge of species to discriminate those potentially present without excluding the presence of unexpected species. This requires the inclusion of phylogenetically related (evolutionarily close) species and all the species that could be justifiably found in place of the targeted species. One of the most widely employed approaches in wildlife forensics takes advantage of species-specific primers-PCR, which allow the amplification of a DNA target in a metagenomic sample, then used for the identification at the species level [111]. An example of the importance of identifying a biological species with certainty is represented by the fight carried out by the Indonesian scientific community against the decline of Sumatran tigers (*Panthera tigris sumatrae*) populations caused by intense poaching and trafficking [117]. In this study, 20 biological samples seized by police authorities (first documented case), including hair, blood, and claws suspected to belong to specimens of Sumatran tigers, were sequenced for the COI region, which was confirmed to be a very powerful genetic marker to ascertain that those samples belonged to *P. tigris sumatrae*, thus helping the Indonesian justice to identify different cases of crime [117]. Recently, a combination of multiple minisatellite analysis and wide-genome SNP genotyping was used to solve a case of suspected poaching of wild goats (*Capra caucasica*) in the Caucasus Mountains region of Russia, creating a precedent in the history of Russian wildlife forensics [118].

### 6.2. Identification of Sample Geographical Origin

Wildlife legislation usually operates within political boundaries, such as national and regional borders or marine fishing grounds; however, the distribution of species is influenced by biological and environmental factors and rarely finds concordance with existing laws. This mismatch often leads to investigations concerning wildlife crimes which frequently involve analysis of the geographic origin of threatened species [111]. From a forensic perspective, determining the geographic origin of a specimen is equivalent to establishing its breeding population of origin. Biological populations exhibit varying levels of genetic variation, from extended families to subspecies, making them difficult to be defined. Populations often share genetic material; thus, genetic markers are less useful in defining discrete differences between groups [111]. The identification of geographical origin is based on the ability to assign one sample to a particular population and rely on the existence of population data from multiple areas. Within some species, populations may be isolated from others, and they could not be subject to gene flow. In this scenario, genetic differences are accumulated over evolutionary time until members of a population living in an isolated region will display similar types of genetic markers ([119,120], as examples).

Markers showing such discrete variations are extremely useful in identifying a population and thus applicable in determining the geographic origin of a given individual with a high confidence interval [111]. The hypervariable control region of mtDNA is often used as a marker to identify the geographic origin, with individual types of control region sequences (haplotypes) corresponding to specific populations. In the event of low mitochondrial variability, it is necessary to analyze nuclear DNA markers, which display a genetic variation between regions. Although some minisatellite and SNP markers show discrete differences, individual alleles are often distributed across populations; thus, the degree of differentiation can be assessed only through different allele frequencies in order to characterize the genetic structure and estimate the probability of a sample belonging to a certain geographic area [111].

This probabilistic approach has two important implications in a wildlife forensic investigation: (1) a large genetic database representative of the allele frequencies of all potential source populations must be set up; (2) statistical analyses are needed to provide a quantitative value of the probability of assigning the sample to a potential population [110].

### 6.3. Identification of Individuals

Over the past two decades, DNA profiling and phenotyping for individual identification was strengthened by the institution of the VISAGE (Visible Attributes Through Genomics) Consortium [121,122,123] and has revolutionized human forensic analysis. In cases such as poaching, it can produce damning evidence by supporting to prove that a horn, tusk, bone, or skin comes from a specific individual. Other applications also include the identification of stolen animals and authentication of legally traded wildlife products [3,4,5,111]. DNA profiling consists of targeting genetic markers (microsatellites or SNPs) that are highly variable within species and can be excellent tools for highlighting genetic differences between individuals [16]. However, when two samples share the same DNA profile, it can be assumed that they come from the same individual; thus, it will be necessary to calculate the probability with which two individuals may randomly have the same profile. This value is influenced by the number and the variability of markers, the allele frequencies within a species, and the relationships among individuals belonging to the same population. The evaluation of these factors needs a representative sample of DNA profiles from the population [111]. Analysis of individual DNA profiles can also be used to regulate the legal trade of species that enjoy protection only in certain parts of their areal or are subject to restricted quotas [15,16]. Wildlife DNA registries, in which legally traded specimens can be individually recognized via their DNA profile, represents the way to ensure that illegal wildlife captured cannot be traded via the supply chain [15,16,111]. An interesting example of such an application is represented by a study focused on the development of an updated approach for elephant genotyping [124]. Researchers performed a multiplex genotyping assay aiming to ascertain a method for the identification of African (*Loxodonta africana*) and Asian (*Elephas maximum*) elephants by the individuation of six tetranucleotide STRs and two sex typing markers in both taxonomic genera [124].

### 6.4. Family Identification

The ability to accept or reject a family relationship represents the final application of wildlife DNA forensic techniques [125]. Defining kinship relationships is crucial both in conservation and forensic genetics, with the main application in the discrimination of wild animals from those bred in captivity. Captive breeding programs of both highly endangered species and animal or vegetable products with a commercial value are currently being carried out all over the world; an example is represented by the trafficking of corals [54,126], parrots, birds of prey, tortoises, and orchids [111]. Obviously, many mammals are also threatened, as occurs in Indian tiger populations [127] or in African cheetahs (*Acinonyx jubatus*) populations, for which a set of SNP markers for individual identification and parentage verification were recently developed in order to monitor the legal and illegal trafficking of cheetahs in South Africa [128]. Since genetic markers are inherited from one generation to the next, DNA profiles allow us to verify the parent–offspring relationship. If alleles observed in a sample do not match those of a putative parental DNA profile, the possibility that the test subject represents his offspring can be excluded. This exclusion method to reject a parentage claim does not require profile data from the wider population and is consequently easier to be performed [111]. However, the variability at a genetic marker level is represented by the occurrence of an inheritable mutational event, where one allele changes to another between generations.

These events can create the possibility of a disagreement between parent and offspring profiles; thus, as done in human genetics, the mutation rate of each marker should be included in the parentage analysis. Unfortunately, this is rarely possible for wildlife species, and the interpretation of genetic profiles requires more attention [111].

## 7. Towards Innovative Technologies

The advent of new molecular biology techniques has facilitated the resolution of many forensic cases otherwise unsolvable through classical approaches (morphometric and behavioral analysis, etc.). Currently, genetic markers and specific regions targeted by the Sanger sequencing method are largely employed to corroborate and confirm morphological evidence for species identification and/or phylogenetic reconstructions. The advent of new genomic technologies based on massive parallel sequencing (MPS) has provided faster and more informative analysis of genomic variation, such as high-resolution genotyping and copy number variation (CNV). As reviewed by [129,130,131], we are at the beginning of the forensic genomics era in which the MPS platforms are steadily introduced in forensic laboratories. Industries such as Verogen, Illumina, Thermo Fisher Scientific, and Promega have developed and commercialized specific human marker panels (e.g., ForenSeq DNA signature prep kit on the MiSeq FGx™ platform, Precision ID Identity and Ancestry Panels on the Ion S5™ platform and PowerSeq™ system by Promega), but non-human marker panels are still not validated for forensic applications.

The continuous growth of available full genomes from different breeds and animal species, mapped and published in open-source databases, opens new opportunities for forensic sciences and implies the extant transition from animal forensic genetic to genomic casework. As animal genomics is a sub-discipline within the larger field of forensic genomics, it takes advantage of MPS breakthroughs on human samples with parallel principles and approaches, but it shows different limitations and needs that represent new challenges:Reference genomes and population reference databases;As much as possible variation analyzed in a unique experiment;Untargeted approaches for unknown and multiple samples;Quality control, assurance, and certification: protocol development based on genomic big data and population genomics, and consequent standardization;Transitioning forensic DNA analysis from the laboratory to the crime scene or on the field with single-molecule sequencing methods;Joint efforts and concerted protocols between scientific communities (primarily the International Society for Animal Genetics, International Society for Forensic Genetics, Society for Wildlife Forensic Science, and International Society of Environmental Forensics) in order to establish shared recommendations, standards, and guidelines.

The emergence of third-generation sequencers, especially Oxford Nanopore Technologies (Oxford, UK) with the MinION sequencer, has proven the possibility of obtaining sequences of reliable quality even in remote areas or under adverse environmental conditions. Being small and powered via USB, it can be used directly on the field, thus eliminating the transport of biological samples and drastically lowering DNA degradation, with consequent improvement of quality and reliability of genomic analyses [132,133].

Even if Oxford Nanopore Technologies suffers from low reading accuracy when compared with Sanger and Illumina technologies, especially in homopolymeric regions preserves great potential if applied to wildlife forensic analyses [134]. Very often, countries richest in biodiversity are those with the worst ‘economic health’ and are characterized by the absence of research infrastructures capable of supporting the protection of the territory and species victims of poaching and illegal trafficking through scientific documentation. The use of nanopore technology sequencers could therefore represent the key to overcoming technical limitations imposed by economic constraints.

Notwithstanding, MinION-based sequencing has been claimed to be not fully usable for profiling STR sequences since the high frequency of homopolymeric regions, and the short length [135], errors produced during the STR typing are strictly dependent on the locus examined [136]. A similar result was achieved by Ren and colleagues in a study focused on the efficacy of genetic marker profiling based on nanopore technology; they were able to identify 94 SNP and 32 STR loci reliably typable by MinION sequencing [137]. Additionally, if this sequencing method turns out to be more error-prone than the profiling of SNPs [137], reads obtained with nanopore technology can be employed to reach forensic STR profiles with high accuracy [138]. In conclusion, the rapid advancement of Oxford Nanopore Technologies instrumentation and the growing scientific literature focused on analyzing and improving the read reliability of nanopore sequencing technology on different genetic loci seem to indicate a gradual replacement of current sequencing technologies, at least in the faunal applications of forensic science.

## 8. Conclusions and Perspectives

Unlike human genetic forensics, the field of animal DNA forensics still suffers from a lack of agreed analytical approaches and standardized procedures. Nevertheless, the growing connection between human and animal habits has led the latter to become ‘eyewitnesses’, very often incriminating. Thus, the use of biological material belonging to animals has now become procedural in many forensic investigations concerning a wide range of crimes, changing the past anthropocentric approach of forensic sciences. As a result, even the protocols followed by animal genetics laboratories have undergone improvements and refinements in order to guarantee the maximum reliability of the genetic evidence presented in a court of justice. Recently, the breakthrough in human genomic resolution has been successfully applied to animal DNA mostly from domestic animals, resulting in extreme legal importance, especially in investigations that are difficult to be solved. However, forensic animal genomics does not deal only with crimes perpetrated between humans or by animals on humans; there is a different branch, the so-called wildlife forensic, which deals with the conservation of biodiversity and the protection of endangered species, contrasting poaching and illegal trade. Many approaches standardized for human DNA are also used in a faunal application of forensic science, but the intervention place is often impervious and makes it difficult to guarantee good preservation of samples. Fortunately, new possibilities have glimmered with the development of third-generation sequencing technologies, such as Oxford Nanopore Technologies, a unique scalable technology that enables field, direct, and real-time sequencing, thus drastically reducing times and costs.

## Figures and Tables

**Figure 1 ijms-24-08821-f001:**
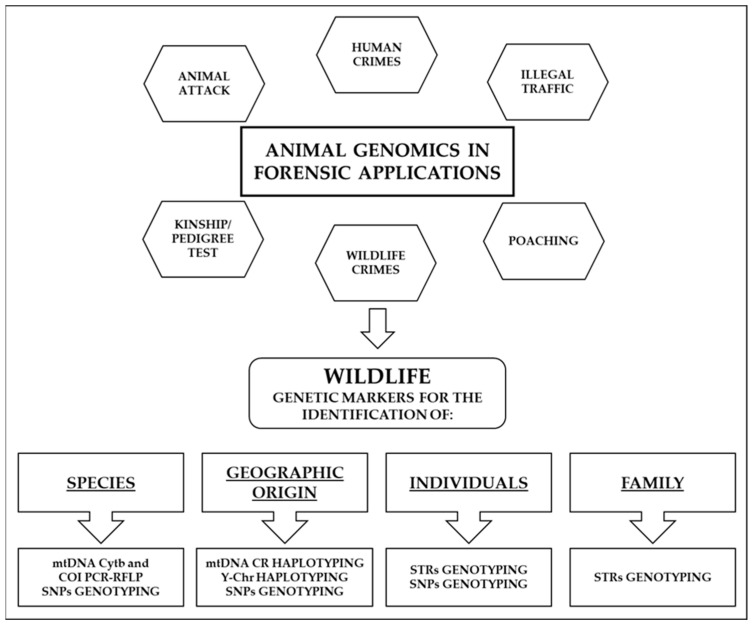
Summary of the most frequent applications of genetic analysis in wildlife forensics.

**Table 1 ijms-24-08821-t001:** Summary of the main animal platforms and databases (available on May 2023).

Database	Website	Description
ADW (Animal Diversity Web)	https://animaldiversity.org/	Online database of animal natural history, distribution, classification, and conservation biology
Avibase (The World Bird Database)	https://avibase.bsc-eoc.org/	Extensive database information system about all birds of the world
Barcode of Life Data System	https://www.boldsystems.org/	Online workbench and database of DNA barcode data
CABI (Commonwealth Agricultural Bureaux International)	https://www.cabi.org/product-training/online-resources/	Bibliographic Database
DDBJ (DNA Data Bank of Japan)	https://www.ddbj.nig.ac.jp/	DNA Database
EMBL (European Molecular Biology Laboratory)	https://www.embl.org/	Intergovernmental organization focused on molecular biology
FAO (Food and Agriculture Organization)	https://www.fao.org/home/en/	Specialized agency of the United Nations
Fauna Europaea	https://fauna-eu.org/	Main zoological taxonomic index in Europe
FishBase	https://www.fishbase.se/	Database on the biology of fish
ForCyt Project	https://www.ForCyt.org	Fully regulated database of species encountered in forensic investigations
GenBank	https://www.ncbi.nlm.nih.gov/genbank/	Genetic Sequence Database in NCBI (National Center for Biotechnology Information)
IBAT (Integrated Biodiversity Assessment Tool)	https://www.ibat-alliance.org/	Database of the global biodiversity
Loxodonta Localizer	https://www.loxodontalocalizer.org/	Tool for inferring the provenance of African Elephants and their ivory using mtDNA
NatureServe	https://www.natureserve.org/	Source for conservation science and biodiversity data in North America
RhODIS (Rhino DNA Index System)	http://rhodis.co.za/	Store of the genetic fingerprint of every rhino that has been sampled
Systema Naturae	https://systemanaturae.org/	Database for the protection of biodiversity
Wild Welfare	https://wildwelfare.org/	Online zoo animal databases and associations for the wild welfare
WoRMS (World Register of Marine Species)	https://www.marinespecies.org/index.php	List of names of marine organisms, including information on synonymy
Zoonomia Project	https://zoonomiaproject.org/	International collaboration to discover the genomic basis of shared and specialized traits in mammals

**Table 2 ijms-24-08821-t002:** Summary of main applications/advantages and recommendations/limitations of genetic markers used for animal forensic analyses.

Genetic Marker	Applications/ Advantages	Limitations/ Recommendations	References
STR	Individual/population identification and kinship testing; many characterized species	Lack of representative samples (wild species) and high-quality genomic sequences; presence of artifacts; integrity and traceability of sample collection	[8,75,78,79]
Y-chromosome STR	Gender identification, resolving paternity and family structures	Uncharacterized for many animal species	[75,78]
X-chromosome STR	Gender identification, resolving paternity and family structures	Uncharacterized for many animal species	[75,78]
Autosomal SNPs	Individual identification	Highly degraded or low template DNA	[75,78]
Mitochondrial DNA	Species identification; high copy number per cell; useful in degraded samples	Not used for individual or breed identification	[75,78]

**Table 3 ijms-24-08821-t003:** Summary of main applications/advantages and recommendations/limitations of genetic markers used in human forensic analyses.

Genetic Marker	Applications/ Advantages	Limitations/ Recommendations	References
STR	Technical robustness and high variation among individuals; several commercial kits; high level of discrimination, standardized across laboratories	PCR artifacts; highly degraded or low template DNA; certain degree of linkage between some STR markers	[8,80,81,82]
Y-chromosome STR	Gender identification, resolving paternity and family structures	Database sufficiently large and continuously expanded; samples need to be collected randomly; each haplotype submission must include metadata	[83]
X-chromosome STR	Gender identification, resolving paternity and family structures	Evaluation of DNA mixture profiles and linked markers	[81]
Autosomal SNPs	Individual identification; information about physical traits	Highly degraded or low template DNA	[81]
Mitochondrial DNA	High copy number per cell; useful in degraded samples; existence of rich population databases; software tools for phylogenetic checks and data quality control	Inter-laboratory differences about the interpretation of length and point heteroplasmy; not used for individual or breed identification	[60]

## Data Availability

No new data were analyzed in this study. Data sharing is not applicable to this article.

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
