# Peer review of "The Revolution of Animal Genomics in Forensic Sciences"

_ijms, 2023, doi:10.3390/ijms24108821_

Round 1
Reviewer 1 Report
In abstract part: Rewrite, it's not acceptable in this form.
Introduction part: Brief note on Forensic sciences focus on animal genetics with latest references and also make a clear table and a clear figure and explain them in text. Figure must be complex and must clearly explain the role of the revolution of animal genomics in forensic sciences.
MAKE A LIST OF population reference databases in table form.
make a list of the online websites, where people can search out animal related genomics study.
References: Must b MDPI format and appropriate and adequate reference to related and previous work added.
Plagiarism must be less than 15%
Nill
Author Response
In abstract part: Rewrite, it's not acceptable in this form.
- Authors’ response: we would thank the reviewer for his suggestion. In accordance to the journal guidelines, we have modified the abstract as follows: “Nowadays, the coexistence between humans and domestic animals (especially dogs and cats) has become a common scenario of daily life. Consequently, during a forensic investigation in civil or criminal cases, the biological material from a domestic animal could be considered as “evidence” by law enforcement agencies. Animal genomics offers an important contribute in cases of attacks and episodes of property destruction, or in a crime scene where the non-human biological material is linked to the victim or perpetrator. However, only few animal genetics laboratories in the world are able to carry out a valid forensic analysis, adhering to standards and guidelines that ensure the admissibility of data before a court of law. Today, forensic sciences focus on animal genetics considering all domestic species through the analysis of STRs (Short Tandem Repeats), autosomal and mitochondrial DNA SNPs (Single Nucleotide Polymorphisms). But the application of these molecular markers to wildlife seems to have gradually gained a strong relevance, aiming to tackle the illegal traffic, avoid the loss of biodiversity, and protect endangered species. The development of third-generation sequencing technologies has glimmered new possibilities, by bringing "the laboratory into the field", with a reduction of both the enormous cost management of samples and the degradation of the biological material.”
Introduction part: Brief note on Forensic sciences focus on animal genetics with latest references and also make a clear table and a clear figure and explain them in text. Figure must be complex and must clearly explain the role of the revolution of animal genomics in forensic sciences. MAKE A LIST OF population reference databases in table form. Make a list of the online websites, where people can search out animal related genomics study.
- Authors’ response: as suggested by the reviewer, we have updated the Introduction part with latest references and provided a list of reference databases and online websites in a table form.
References: Must b MDPI format and appropriate and adequate reference to related and previous work added.
- Authors’ response: we would thank the reviewer for the observation. References follow MDPI guidelines and have been formatted by using the Zotero tool.
Plagiarism must be less than 15%
- Authors’ response: we would thank the reviewer for the observation. We used the Compilatio Magister software (www.compilatio.net/it/studium) as an anti-plagiarism tool to value academic authenticity. The whole manuscript was checked and the resulting percentage of similarities was equal to >1% (the certificate report is available if requested), thus in accordance to the editorial guidelines.
Reviewer 2 Report
A brief summary
The manuscript is interesting and smoothly read. But I think you need to talk about the crimes to animals by humans not only the attack of animals on humans and the role of animals in human crimes. Also, you specified dogs only from all animal species. In addition to animal behavioural genetic analysis role in forensic science which you just mentioned but not add evidence to it from literature.
Specific comments
Abstract
- Line 17-18: remove (above) before (all), write (considering) not (concerning)
- Line 19: stat a new line with (but)
- Line 23: start a new line with (the development)
Keywords
- Write (domestic) not (domestics)
Introduction
- Line 55-56: not clear, clarify
- Line 57-58: remove (can) before (consider), write (both) not (two branches), remove (species)
- Line 59: add (biological material) before (are)
Animal DNA in human cases
- Line 65-66: repeatable
- Line 74: add (that was) before (used)
- Line 96-98: not clear why not use nuclear DNA, clarify
- Line 100: write (also) not (anyway)
Recommendations for animal DNA analysis in forensic sciences
- Line 180: add (which) before (the)
- Line 184: add (will be) before (more)
Domestic animal DNA as forensic evidence
- Line 267: not always dog have dense hair as there is some skinny breeds, also cats have more hair than dogs. I think you need to mention that they have more hair than human or state that the crime scene can contain mor hair form dog than human.
- Line 348-350: mention the reference for this statement as it is confusing
Wildlife in forensic genetics
- Figure 1: remove the green color as not clear writing
- Line 509: add (was) before (strengthened)
- Line 510-513: too long, shorten to clarify
Minor editing
Author Response
A brief summary
The manuscript is interesting and smoothly read. But I think you need to talk about the crimes to animals by humans not only the attack of animals on humans and the role of animals in human crimes. Also, you specified dogs only from all animal species. In addition to animal behavioural genetic analysis role in forensic science which you just mentioned but not add evidence to it from literature.
- Authors’ response: we would thank the reviewer for his observation. We have now improved the manuscript as suggested, and we have modified the title of the second chapter in “Animal DNA in forensic cases”.
Specific comments
Abstract
- Line 17-18: remove (above) before (all), write (considering) not (concerning)
- Authors’ response: we have modified the text as suggested.
- Line 19: stat a new line with (but)
- Authors’ response: we have modified the text as suggested.
- Line 23: start a new line with (the development)
- Authors’ response: we have modified the text as suggested.
Keywords
- Write (domestic) not (domestics)
- Authors’ response: we have modified the keyword as suggested.
Introduction
- Line 55-56: not clear, clarify
- Authors’ response: we would thank the reviewer for this remark. We have deleted the repetition “to legal or law enforcement issues concerning animal biological material”
- Line 57-58: remove (can) before (consider), write (both) not (two branches), remove (species)
- Authors’ response: we have modified the text as suggested.
- Line 59: add (biological material) before (are)
- Authors’ response: we have modified the text as suggested.
Animal DNA in human cases
- Line 65-66: repeatable
- Authors’ response: we would thank the reviewer for his suggestion. We have deleted this sentence.
- Line 74: add (that was) before (used)
- Authors’ response: we have modified the text as suggested.
- Line 96-98: not clear why not use nuclear DNA, clarify
- Authors’ response: we have modified the sentence as follows: “A hair without root usually contains little nuclear DNA, thus, in that case it was not possible to generate a nuclear DNA profile. Therefore, mtDNA was analyzed and its classification in one haplotype typical of only 9% of the world's canine population, including the victim's dog, was used as evidence to link Westerfield to the murder of the little girl, and to obtain his conviction [17].”
- Line 100: write (also) not (anyway)
- Authors’ response: we have modified the text as suggested.
Recommendations for animal DNA analysis in forensic sciences
- Line 180: add (which) before (the)
- Authors’ response: we have added “which are” before “the”.
- Line 184: add (will be) before (more)
- Authors’ response: we have modified the text as suggested.
Domestic animal DNA as forensic evidence
- Line 267: not always dog have dense hair as there is some skinny breeds, also cats have more hair than dogs. I think you need to mention that they have more hair than human or state that the crime scene can contain mor hair form dog than human.
- Authors’ response: we would thank the reviewer for his suggestion. We have modified the text as follows: “In addition, since dogs have more hair than human, thus the crime scene can contain more hair from dog than human, the probability of finding canine-derived hair is higher compared to recovery traces of human biological material”.
- Line 348-350: mention the reference for this statement as it is confusing
- Authors’ response: we have added the reference and modified the text as follows: “Attacks by domestic dogs usually aim to head and neck regions, whereas those by stray dogs often affect hands and legs, probably due to the different human behavior towards domestic and wild dogs [Fonseca et al. 2015].”
Wildlife in forensic genetics
- Figure 1: remove the green color as not clear writing
- Authors’ response: we have modified the figure as suggested.
- Line 509: add (was) before (strengthened)
- Authors’ response: we have added “was” before “strengthened”.
- Line 510-513: too long, shorten to clarify
- Authors’ response: we have shortened the sentence as follows: “Over the past two decades, DNA profiling and phenotyping for individual identification was strengthened by the institution of VISAGE (Visible Attributes Through Genomics) Consortium [89–91], and has revolutionized human forensic analysis. In cases such as poaching, it can produce damning evidence by supporting to prove that a horn, tusk, bone, or skin come from a specific individual.”
Reviewer 3 Report
A review article in line with current trends in interdisciplinary research. A certain drawback is the authors' focus on the domestic dog in the chapter on domestic animals. According to FEDIAF, now the domestic cat is the most popular pet in Europe.
Small mammals such as rabbits, ferrets and hamsters are becoming more and more popular as companion animals. It would be good to at least mention these species by citing one or two scientific articles devoted to them.
Farm animals are another important group of animals from the point of view of breeding and economy. There is no mention of, for example, adulteration in products of animal origin. In my opinion, a few sentences with citing the literature should also be added.
Finally, it is necessary to describe the use of molecular biology methods in the care of the purity of breeding lines of laboratory animals.
Author Response
A review article in line with current trends in interdisciplinary research. A certain drawback is the authors' focus on the domestic dog in the chapter on domestic animals. According to FEDIAF, now the domestic cat is the most popular pet in Europe. Small mammals such as rabbits, ferrets and hamsters are becoming more and more popular as companion animals. It would be good to at least mention these species by citing one or two scientific articles devoted to them.
- Authors’ response: we would thank the reviewer for his suggestion. We have now improved the manuscript by mentioning also other companion animals.
Farm animals are another important group of animals from the point of view of breeding and economy. There is no mention of, for example, adulteration in products of animal origin. In my opinion, a few sentences with citing the literature should also be added.
- Authors’ response: we would thank the reviewer for his suggestion. We have now improved the manuscript by including also farm animals and their importance in the agri-food sector.
Finally, it is necessary to describe the use of molecular biology methods in the care of the purity of breeding lines of laboratory animals.
- Authors’ response: we would thank the reviewer for his suggestion. We have added the missing information as follows: “The development of genetic markers finds an important application also in the analysis of laboratory animals, by ensuring the purity of breeding lines, a good monitoring of model species, and the identification and variation of genes acting on specific traits [Johnsson et al. 2023; Sin et al. 2023; Yoshiki et al. 2022; Benavides et al. 2020; Flint and Woolliams, 2008]”.
Round 2
Reviewer 1 Report
nill